# Natural Receptor- and Ligand-Based Chimeric Antigen Receptors: Strategies Using Natural Ligands and Receptors for Targeted Cell Killing

**DOI:** 10.3390/cells11010021

**Published:** 2021-12-22

**Authors:** Gianna M. Branella, Harold Trent Spencer

**Affiliations:** 1Aflac Cancer and Blood Disorders Center, Department of Pediatrics, School of Medicine, Emory University, Atlanta, GA 30322, USA; gianna.branella@emory.edu; 2Cancer Biology Program, Graduate Division of Biological and Biomedical Sciences, School of Medicine, Emory University, Atlanta, GA 30322, USA; 3Marcus Center for Pediatric Cellular Therapy, Children’s Healthcare of Atlanta, Atlanta, GA 30322, USA

**Keywords:** chimeric antigen receptor, CAR, ligand-based CAR, natural receptor-based CAR, immunotherapy, cell therapy, CAR T cell therapy

## Abstract

Chimeric antigen receptor (CAR) T-cell therapy has been widely successful in the treatment of B-cell malignancies, including B-cell lymphoma, mantle cell lymphoma, and multiple myeloma; and three generations of CAR designs have led to effective FDA approved therapeutics. Traditionally, CAR antigen specificity is derived from a monoclonal antibody where the variable heavy (V_H_) and variable light (V_L_) chains are connected by a peptide linker to form a single-chain variable fragment (scFv). While this provides a level of antigen specificity parallel to that of an antibody and has shown great success in the clinic, this design is not universally successful. For instance, issues of stability, immunogenicity, and antigen escape hinder the translational application of some CARs. As an alternative, natural receptor- or ligand-based designs may prove advantageous in some circumstances compared to scFv-based designs. Herein, the advantages and disadvantages of scFv-based and natural receptor- or ligand-based CAR designs are discussed. In addition, several translational aspects of natural receptor- and ligand-based CAR approaches that are being investigated in preclinical and clinical studies will be examined.

## 1. Introduction

Chimeric antigen receptor (CAR) T cells were first described in the late 1980s as an indirect method to determine the differences in antigen recognition between T cells and B cells [1], and later as a means of redirecting the activation and effector function of patient-derived T-cells toward a target cell via antibody-specific antigen selectivity [2,3,4]. In this context, T cells do not require antigen processing and presentation by the major histocompatibility complex (MHC) and can, therefore, elicit an immune response against any cell, traditionally those that are cancerous or virally infected. CAR T therapy has expanded to the treatment of many cancer types with the greatest success in the treatment of CD19+ B-cell malignancies, where the first two CAR products became approved by the United States Food and Drug Administration (FDA) in 2017 [5,6,7]. More recently, three additional CAR T therapies have been approved for relapsed or refractory mantle cell lymphoma, B-cell lymphoma, and multiple myeloma [8,9,10].

CARs have a modular design that allows for disease-application specific customization. As shown in Figure 1, they broadly consist of an antigen-binding domain, a hinge, a transmembrane domain, and an intracellular signaling domain [11,12]. The antigen-binding domain has traditionally included combinations of the variable heavy (V_H_) and variable light (V_L_) chains of a monoclonal antibody connected by a short peptide linker to form a single-chain variable fragment (scFv). Connected to the scFv is the hinge region (or sometimes called a “spacer” region), which promotes flexibility within the CAR construct and has been shown to affect the persistence and efficacy of CAR T cells [13,14,15,16]. Following the hinge region is the transmembrane domain, which can exist as part of the hinge or costimulatory domain. Finally, the intracellular signaling domain provides the cytotoxic activity of the T cell and, as such, consists of one or two costimulatory domains (typically either CD28, 4-1BB, ICOS, or OX-40) and a primary signaling domain, CD3ζ. Together, the parts of this chimeric cell surface protein function in unison to promote customizable and specific cell killing.

All CAR T therapies currently approved by the FDA utilize an scFv as the antigen-recognition domain of the CAR construct. As they are derived from antibody sequences, scFvs have a unique ability to bind their target antigen with high specificity and affinity. However, this design is not without flaws. Primarily, scFvs hold endless possibilities for target antigen specificity given one large stipulation: there exists a monoclonal antibody for the given receptor of choice. While this is not an issue for more prevalent targets of choice, this may be an issue for nuanced target receptors that have not been as extensively investigated. However, it is important to state that this is not always a limitation of the technology, as an investigator may develop a new antibody and scFv sequence for their studies. Beyond this, scFv instability (influenced by external factors such as temperature and pH) [17] or structurally intrinsic factors [18] can encourage protein unfolding or scFv aggregation through domain swapping [19,20,21], which may translate to inferior clinical efficacy. Further, issues surrounding scFv immunogenicity [22] can require discontinuation or CAR T elimination in some instances, which can also lead to inferior in vivo function [23,24].

Alternative to the traditional scFv-based design, natural receptor- and ligand-based CARs utilize the naturally occurring specificity of a receptor or ligand as their antigen specificity and can therefore capitalize on naturally evolved ligand-receptor interactions—given there exists a natural ligand to the targeted receptor—as demonstrated in Figure 1 (top left). 

In contrast to the scFv-based CARs (Figure 2A), natural receptor-based CARs maintain the ectodomain (and sometimes the transmembrane domain and/or signaling domain) of the receptor of choice (Figure 2B), followed by the traditional modules of a CAR, whereas a ligand-based CAR simply replaces the scFv portion of an scFv-based CAR with the sequence for a natural ligand (Figure 2C). Both the natural receptor and natural ligand of these alternative CAR designs maintain the natural affinity to their binding partners, which allows for target cell-specific binding and initiation of downstream effector CAR T function.

Like the use of scFv-based CARs, natural receptor- and ligand-based CARs have their advantages but also have their own concerns. In this review, the advantages of natural receptor- and ligand-based CARs are described in the following section, followed by a discussion of the challenges and drawbacks of these alternative CAR designs. A review of the current natural receptor- and ligand-based CAR T cells currently in preclinical and clinical development follows.

## 2. Disadvantages of scFv-Based CARs in Comparison to Natural Receptor-/Ligand-Based CAR T Cells

There are some advantages to a natural receptor- or ligand-based CAR design over a traditional scFv-based design. As described below, these advantages center on resolving known disadvantages of scFv-based constructs (Figure 3), mainly the issue of scFv instability leading to CAR aggregation, which can induce tonic signaling and low in vivo function.

### 2.1. scFv Instability and Domain Swapping Leads to scFv Aggregation and Tonic CAR Signaling

Many studies demonstrate the scFv instability through external factors such as temperature or pH changes [17,19,25] (Figure 3H). Drastic temperature changes may not be of great concern during CAR T development, as thawed CAR T cells likely produce new properly folded CAR molecules upon treatment. In contrast, pH changes may be intrinsic to in vivo applications, such as the hypoxic tumor microenvironment (TME). However, sufficient studies have yet to determine such in vivo effects.

Factors intrinsic to the structure of the scFv, such as solvent exposure of hydrophobic residues at the variable-constant region interface or the loss of stabilizing interactions normally found in the constant region of the antibody [18,24], can also have profound effects on scFv stability. To construct an scFv, the V_H_ and V_L_ domains of a monoclonal antibody are taken out of their natural context where they are normally associated with the constant regions of the heavy and light chains of a monoclonal antibody and are instead connected by a linker of varying amino acid sequences and lengths. In doing so, hydrophobic amino acid residues normally situated between the variable and constant regions of an antibody that would not naturally be exposed to solvent now are, which can lead to protein unfolding (Figure 3F). Nieba et al. [18] examined 30 antibody fragment structures and determined that the frequency of solvent-exposed hydrophobic residues at the variable-constant interface was higher in comparison to the rest of the structure. However, these unstable scFv designs are usually rapidly identified and eliminated in preclinical studies due to a lack of in vivo efficacy. Furthermore, some have gone on to clinical trials, such as the GD2 CAR Tcells based on the 14g2a scFv, which have been shown to tonically signal in the absence of antigen due to instability in the scFv framework regions [23].

Further, many studies have demonstrated the oligomerization of scFv molecules, including soluble scFvs [18,25,26,27,28], scFvs fused to constant antibody fragment (Fc) regions [29], and even scFvs as a part of a CAR structure [23,30]. These scFv dimers, trimers, or even tetramers have been coined diabodies [31], miniantibodies [32,33], and multivalent variable fragments (Fvs) [28]. Multiple scFvs primarily oligomerize through domain swapping [20,21], where the V_H_ region of one scFv will incorrectly associate with the V_L_ region of another scFv (Figure 3A). This causes scFv (and therefore CAR) aggregation, which will result in non-functional in vivo efficacy [23,24,25,34,35] and, particularly in the case of CAR T therapy, leads to tonic signaling via the constitutive activation through NF-κB, AKT, ERK, and NFAT signaling [23,30] (Figure 3B).

Tonic signaling in immunocompetent cells has been shown since the late 1990s as a means for T- and B-cells to regulate their survival and function [36,37]. Defined as a low level of constitutive activation in nonactivated basal state T-cells, tonic signaling is mediated by non-antigen-specific interactions with antigen-presenting cells (APCs) [38,39]. In this context, tonic signaling serves as a means to enhance future T-cell reactivity [40]. However, in the context of CAR-modified T-cells, tonic signaling can result in constitutive cytokine release, prolonged expansion, and enhanced T-cell exhaustion. Many scenarios can lead to CAR tonic signaling besides scFv aggregation, including the level of CAR expression, certain costimulatory domains [24], and the addition of endogenous T-cell receptor (TCR) signaling in the CAR T cell [41]. Importantly, tonic signaling observed in the preclinical development of CAR T cells has been shown to enhance the disparity between in vitro and in vivo antitumor efficacy [14,23,24,42].

Long et al. [23] explored the presence of CAR aggregation and tonic signaling through the investigation of a GD2-specific CAR (derived from the 14g2a clone), two CARs directed toward CD22 (HA22 and m971 scFvs), the HER2-targeted CAR (scFv 4D5), and the clinically used CD19 CAR (scFv FMC63). They found that every CAR, with the important exception of the CD19 CAR, experienced various degrees of tonic signaling. This may explain the great clinical success of the CD19 CAR. Further, they demonstrate that GD2 CAR tonic signaling can be abrogated upon the replacement of GD2 antibody framework regions with the CD19 antibody framework regions, pinpointing the source of scFv aggregation to the framework of the scFv as opposed to the complementarity determining regions (CDRs). Others have shown similar framework-replacement methods to improving scFv stability, soluble scFv yield, function, and thermodynamic stability by decreasing aggregation [43,44].

Importantly, CAR aggregation likely inhibits the recognition of the target antigen (Figure 3C,G), which negatively affects in vivo function [23,24]. Furthermore, due to conformational changes in the overall structure, scFv aggregates may lead to immunogenicity concerns due to enhanced conformational changes in the scFv structure, leading to recognition by the host immune system [17,34,35] (Figure 3D), though more studies are needed.

In comparison, natural ligands and receptors are likely more stable and have decreased dimerization and domain swapping risk, apart from some multimeric ligands. However, multimeric ligands pose little risk of immunogenicity as they are naturally occurring oligomerizing interactions, though more studies are needed. Further, an important question to consider is the influence of a native TCR-like promoter in the generation of CAR T cells, as this alone may assuage the issue of tonic signaling rather than investing the time in developing a natural ligand-based CAR.

### 2.2. scFv Immunogenicity

Many studies have demonstrated the immunogenicity of murine or humanized scFvs [25,45], including one early clinical trial showing both humoral and anti-CAR T cell immune responses upon treatment with a murine-derived anti-CAIX CAR resulting in low CAR T function and persistence [22]. The clinically-successful anti-CD19 scFv FMC63 is also murine-derived, though fully human versions have been shown to be superior [46].

Roque et al. [47] and Wang et al. [17] review the immunogenicity of various antibody structures. In summary, murine-derived antibodies commonly undergo a rigorous humanization process that replaces all portions of the murine antibody using human counterparts except for the murine CDR regions, which are responsible for antigen recognition, and result in an antibody that is only 5–10% murine-derived. While these highly humanized antibodies have nearly the same immunogenic potential as fully human antibodies, there is still a small chance to produce human anti-human antibodies (HAHA) upon treatment. However, rare and usually not clinically severe, the development of HAHA to fully humanized antibodies has been shown in the clinic [48] (Figure 3E). That said, it is important to consider that CAR T therapy is administered after immunosuppression, whereas therapeutic antibodies are not. Therefore, scFv-based or natural receptor-/ligand-based CAR T cells may be less immunogenic simply due to the way they are administered.

In contrast to scFvs, natural ligands and receptors do not pose a risk of immunogenicity as they are fully human versions of the protein without any percentage of non-human portions. However, it is important to note that ligand-based CARs (and in some cases natural receptor-based CARs) may still pose a greater risk of immunogenicity than most natural receptor-based CARs due to the necessity of a hinge/transmembrane region, where the interface between the two can initiate an immune response.

Additionally, it is also important to consider allelic variants of natural receptors or ligands, where a certain allelic variant may be immunogenic in a particular population. Further investigation is needed to determine the consequences of such a design.

### 2.3. Antigen Escape 

While efficacious, approximately 7–25% of patients who respond to treatment with CD19 CAR T cells in phase I and II clinical trials have been shown to relapse with CD19-negative disease [5,7,49,50,51,52,53,54,55] in a process known as antigen escape. 

Antigen escape has been observed in at least one clinical trial evaluating natural receptor- and ligand-based CAR T cells [56]. However, the design of natural receptor- and ligand-based CARs may offer greater potential for solutions to this problem. Primarily, the predetermined affinity that receptors and ligands have for multiple binding partners may prove advantageous, as seen in the case of the TriPRIL CAR T cells that dually target two receptors commonly co-expressed in multiple myeloma, B-cell maturation antigen (BCMA) and transmembrane activator and calcium-modulator and cyclophilin ligand (TACI) [57]. Although the issue of antigen escape is a known concern when using scFv-based designs with respect to ligand-based CARs, there is little information regarding this phenomenon. Further, greater binding partners may enhance the risk of on-target, off-tumor toxicities, and therefore need to be carefully examined. Additional research is needed to address this potential concern.

### 2.4. Target Recognition through Evolved Ligand Affinity vs. Antibody Directed scFv Affinity 

The association between scFv affinity and CAR function has been well studied, with early literature suggesting a positive correlation between the two [58,59]. However, recent research has suggested the use of lower affinity scFvs to limit the amount of cell-to-cell contact [60,61]. This can lead to increased proliferation and cytotoxicity in vitro, enhanced CAR T expansion and antitumor activity in vivo, and enhanced CAR T persistence, and decreased exhaustion, possibly due to less CAR tonic signaling [60,61]. 

While the relative affinities of natural receptors and ligands can also be tuned to optimize CAR functionality in a similar manner to scFvs, the natural affinity that receptors and ligands have for their binding partners may prove to be a better starting point for preclinical CAR design. Indeed, tuning the affinity of one natural ligand-based CAR improved in vivo functionality [62]. However, it is important to note that the level of affinity for the target receptor is highly contextually specific and needs to be optimized for each individual CAR construct. Further, affinity-tuning may enhance the immunogenicity of a natural ligand, and therefore must be considered and properly evaluated.

### 2.5. Ease of Preclinical Development Plans

Natural receptor- and ligand-based CARs offer the advantage of a speedy development process. Upon identification of a target receptor, an investigator may design a natural receptor- or ligand-based CAR with widely accessible protein sequences within a matter of days, proceeding rapidly into preclinical development. In contrast, when designing an scFv-based CAR, one must find and identify the sequence for the humanized monoclonal antibody of choice or—in the case that such does not exist—develop their own humanized monoclonal antibody. After scFv identification, the optimal configuration must be determined for the V_H_ and V_L_ domains and examine various linker sequences to connect the two variable domains. The investigator then must ensure the scFv does not have additional unintended off-target effects as compared to the original monoclonal antibody. This process can be lengthy, depending on the circumstance.

## 3. Challenges of Natural Receptor- and Ligand-Based CARs

While there are many advantages to the use of natural receptor- and ligand-based CARs, these alternative CAR designs come with their own set of limitations. These limitations include an enhanced chance of off-tumor toxicity, interference of endogenous receptor/ligand interactions, and unwanted signaling through the targeted receptor, as described below.

### 3.1. Off-Tumor Toxicity

While the possibly numerous binding partner interactions that natural receptors and ligands possess may decrease the potential for antigen escape, this promiscuous binding could be a drawback of the natural receptor- and ligand-based design if other known or unknown binding partners exist on healthy tissues, leading to on-target yet off-tumor toxicities (Figure 3K). It is important to note that scFv-based CARs also experience on-target, off-tumor toxicities, particularly for CARs that target hematological malignancies other than B-cell leukemias. However, the existence of unknown binding partners that can contribute to toxicity is solely a factor for natural receptor- and ligand-based CAR T cells and must be carefully explored in animal models.

### 3.2. Competition with Endogenous Ligands

Natural receptor- and ligand-based CARs capitalize on the natural protein-protein interaction and therefore bind in the active site of the target receptor. Therefore, a potential major challenge of natural receptor- and ligand-based CARs include the interference of endogenous receptor or ligand binding. In this case, endogenous binding partners may compete for binding with the target receptor with the CAR T therapy, thereby interfering with CAR functionality in vivo (Figure 3I). 

Indeed, co-cultures of MPL-targeted CAR T cells and MPL+ cancer cells supplemented with supraphysiological doses of TPO did interfere with cancer cell cytotoxicity, though specific cell lysis due to MPL-targeted CAR T cells as compared to controls was still observed [63]. This was confirmed in the context of the BCMA/TACI-targeted TriPRIL CARs, where high concentrations of soluble BCMA, TACI, and APRIL added to co-culture experiments only marginally reduced cytotoxicity, and only at the lowest E: T ratios [57].

While it seems competition of endogenous receptors and ligands has only a slightly negative effect on in vitro CAR cytotoxicity, it is important to examine each natural receptor- and ligand-based CAR individually to assess potential risk.

### 3.3. Unintentional Signaling through Targeted Receptor

CAR binding through the endogenous binding site of the target receptor may induce receptor signaling (Figure 3J). Importantly, if the target receptor is known to be involved in cell survival and proliferation, this may induce tumor growth in some cases. Also, shedding of the CAR ectodomain from the CAR T cells may enhance cancer cell proliferation and survival. Indeed, supernatants from MPL-targeted CAR T cells were able to induce canonical MPL signaling in MPL+ cancer cells, as demonstrated by increased levels of pSTAT5 [63]. The level of pSTAT5 was then reversed to basal levels when anti-TPO antibodies were added to MPL+ cells in addition to MPL-targeted CAR supernatants, suggesting that cleaved TPO from the MPL-targeted CAR T cells were the culprit for the enhanced STAT5 signaling [63]. Therefore, plans to reduce CAR shedding should be considered [63].

One solution is to design the ligand so that it can bind to the target receptor but is unable to transduce a signal. Indeed, as described above, the GMR-targeted ligand-based CAR T cells utilize a mutant GM-CSF ligand in this context and observe enhanced anti-tumor functionality [62]. GMR CAR T cells incorporating this mutant GM-CSF design are currently being evaluated in first-in-human clinical trials in Japan (jRCT2033210029).

## 4. Natural Receptor- and Ligand-Based CAR T Cell Therapies in Preclinical Studies

There is a growing body of literature surrounding natural receptor- and ligand-based CAR T cell therapies in preclinical studies, all with varying levels of success and different methods for optimization. Below, and in Table 1, we highlight details of the studies based on the receptors being targeted and focus on their effectiveness in preclinical studies.

### 4.1. B7H6

B7H6 cell surface expression is induced upon cell stress and is present on the surface of some tumor cells while absent on normal, healthy counterparts [64,65]. B7H6 binds the cell surface receptor NKp30, which is commonly expressed on natural killer (NK) cells and a small subset of T-cells, such as gamma delta (γδ) T-cells [64]. This expression pattern prompted the investigation of a natural receptor-based CAR to target B7H6+ tumor cells utilizing the NKp30 receptor ectodomain [66]. In this preclinical study, co-culture of CAR T cells with B7H6+ tumor cell lines produced enhanced interferon-gamma (IFN-γ) secretion and induced significant cytotoxicity.

To optimize the design, Butler et al. [67] utilized directed evolution to develop NKp30 variants that bind to B7H6 with higher affinity than the wild-type receptor. Two high-affinity variants of NKp30 (CC3 and CC5) were better able to bind to tumor cells with low B7H6 expression in comparison to the anti-B7H6 scFv-based CAR (clone TZ47), with faster on/off rates. The variants demonstrated equal cell killing of B7H6^low^ cancer cell lines but had higher interleukin (IL) 2 (IL-2), tumor necrosis factor-alpha (TNF-α), and IFN-γ secretion. Importantly, the high-affinity NKp30 CAR T cells induced lower amounts of IL-6, the pro-inflammatory cytokine that induces cytokine release syndrome (CRS, also referred to as “cytokine storm”) in the clinic (Table 1).

### 4.2. EGFR

Adnectins belong to a family of therapeutic proteins derived from the tenth type III domain of the human fibronectin protein (10Fn3) [115] and are members of the immunoglobulin (Ig) superfamily. As such, adnectins closely resemble the variable domains contained within antibody structures and even contain recognition loops structurally analogous to an antibody’s CDR [116]. Therefore, they have been modified to bind targets other than their natural ligand, and retain high affinity and specificity [117]. In addition, since adnectins are small, single-domain proteins derived from human origin, they are thought to be advantageous to scFv sequences due to decreased potential for immunogenicity and tonic signaling. Epidermal growth factor receptor (EGFR) is commonly overexpressed in many solid tumors and contributes to poor patient prognosis [118]. A comparison of an adnectin-based CAR and an scFv-based CAR showed that adnectin-based EGFR-targeted CARs have similar efficacy in targeting EGFR+ cells when compared to an scFv-based EGFR CAR in both in vitro and in non-small cell lung cancer xenograft murine models [91].

### 4.3. FLT3 

FMS-like tyrosine kinase-3 (FLT3) is one of the most frequently mutated genes in acute myeloid leukemia (AML), accounting for 30% of adult AML cases [119], with internal tandem duplication (ITD) mutations appearing in 24% of adult AML patients [120]. Because FLT3 mutations are associated with poor clinical outcomes [121], Wang et al. [92] developed a ligand-based CAR to target the FLT3 receptor by utilizing the truncated FLT3 ligand (FLT3L). Upon co-culture with FLT3+ AML cell lines and primary AML patient samples, CAR T cells induced specific cytotoxicity, especially in cell lines containing the FLT3 ITD mutation, enhanced degranulation measured by CD107a expression, and improved IFN-γ and TNF-α secretion. Further, treatment with CAR T cells extended the median survival of mice in a xenograft FLT3+ AML mouse model by 40 days compared to control CAR T cell treatment. Importantly, treatment with CAR T cells did not inhibit hematopoietic cell colony formation when cultured with CD34+ healthy human umbilical cord blood stem cells despite FLT3 expression in hematopoietic stem cells (HSCs) [122], indicating little off-target toxicity in the hematopoietic compartment, at least in vitro. Maiorova et al. [93] also developed a ligand-based CAR targeting FLT3 utilizing the full-length FLT3L. The authors found the enhanced killing of FLT3+ THP-1 cell lines by CAR T cells in vitro, though in vivo studies are still needed to determine efficaciousness. 

### 4.4. IL-10R 

Chen et al. [94] designed a ligand-based CAR to target IL10R+ AML utilizing IL-10. Co-culture of CAR T cells with IL-10R+ AML cell lines resulted in CAR T cell activation as shown by upregulation of CD69 and CD25, degranulation by enhanced expression of CD107a, secretion of Granzyme B (GrB), IFN-γ, and TNF-α, and elimination of AML cells in culture. In an AML xenograft murine model, treatment with CAR T cells prolonged the survival of mice in comparison to treatment with control CAR T treatment. Notably, there were no effects on CD34+ umbilical cord blood cells, except for monocytes due to their high IL-10R expression. It is important to note the lack of AML-specific antigens that exist; therefore, off-target effects in the monocyte compartment are to be expected with such a CAR design. 

### 4.5. MPL

CD110, also known as the myeloproliferative leukemia protein (MPL), is overexpressed in many AML patients with expression correlating with faster relapses in patients [123,124]. Thrombopoietin (TPO) is the natural ligand to CD110 and, through receptor binding, facilitates HSC self-renewal and promotes the proliferation and differentiation of megakaryocytes [125,126,127]. To target these hard-to-treat populations, Zoine et al. [63] generated a MPL-directed CAR utilizing the biologically active portion of the TPO protein. In this preclinical study, CAR T cells induced specific cytotoxicity against MPL+ AML cell lines in vitro and in AML xenograft murine models, with on-target, off-tumor toxicity seen in the bone marrow compartment. It can be argued that bone marrow toxicity is advantageous to the model, as most AML patients with high MPL+ tumors go on to receive a bone marrow transplant [128], and treatment with MPL-directed CAR T cells may be useful as a CAR T cell could replace the deleterious side effects of genotoxic pre-transplant conditioning regimens.

### 4.6. IL-11Rα 

Huang et al. [95] verified overexpression of the interleukin-11 (IL-11) receptor-α (IL-11Rα) in osteosarcoma (OS) cell lines, and subsequently designed a ligand-based CAR to target this receptor utilizing the IL-11 peptide. In this preclinical study, they demonstrated that the CAR T cells effectively killed OS cell lines in vitro and inhibited OS lung metastasis in murine models. However, mice treated with control T-cells experienced similar T-cell accumulation in the lung and tumor regression to those treated with IL-11Rα-targeted CAR T cells, suggesting a lack of therapeutic benefit. 

### 4.7. PVR/Nectin-2

CD155, also known as poliovirus receptor (PVR), and poliovirus receptor-related 2, also known as CD112 or nectin-2, are expressed on a variety of tumor tissues, including leukemias [129], ovarian cancer [130], neuroblastoma [131], melanoma [132], and glioblastoma [133]. To target these tumors, Wu et al. [134] developed both first-generation and second-generation DNAM-1 ligand-specific CAR T cells utilizing the extracellular domain of DNAM-1 as the ligand-binding portion. In this case, the addition of costimulatory domains did not enhance CAR functionality, as intratumoral treatment with the first-generation DNAM-1 ligand-targeted CAR T cells was successful in reducing tumor burden in murine xenograft melanoma models. DNAM-1 CAR T cells were also evaluated in the context of gliomas [69] and were found to be successful in prolonging the survival in a human glioma xenograft mouse model. However, PVR is expressed on epithelial cells, endothelial cells, and some immune cells [135], and nectin-2 is expressed on epithelial cells, neurons, and Sertoli cells in the testis [136,137]. This widespread expression may prevent the clinical application of DNAM-1 CAR T cells. 

### 4.8. EPHB4

Ephrin type-B receptor 4 (EPHB4) is widely expressed in many tumor types, including rhabdomyosarcoma (RMS) [138]. Importantly, signal transduction through EPHB4 promotes cell death [139], especially in RMS [140]. Therefore, Kubo et al. [96] developed a ligand-based EPHB4-targeted CAR for the treatment of RMS by utilizing the natural ligand for EPHB4, EPHRIN B2. In this preclinical study, CAR T cells induced cytotoxicity of RMS, OS, and triple-negative breast cancer (TNBC) cell lines even after multiple challenges of cancer cells, with sustained CAR T cell proliferation. In a murine RMS xenograft model, treatment with CAR T cells substantially decreased RMS tumor volume in comparison to treatment with control CAR T cells without a change in weight loss, suggesting low general toxicity. Further, mice treated with CAR T cells had an increase in human CD3+ cells in the peripheral blood in comparison to control CAR T treated mice, suggesting antigen-specific T-cell expansion. This data serves as the foundation for further preclinical analysis of EPHB4-directed CAR T cells in the context of pediatric soft tissue sarcomas, and future clinical trials taking place in Japan are being planned.

## 5. Natural Receptor- and Ligand-Based CAR T Cell Therapies in Early Phase Clinical Testing

Many natural receptor- and ligand-based CAR T cell therapies are moving into clinical testing in the US and Japan, though most are currently recruiting and therefore do not have data yet. Below, and in Table 1, we describe the preclinical studies of these non-traditional CAR T cell therapies and provide an update on their clinical trial design and any early signs of efficacy.

### 5.1. NKG2D Ligands

The NKG2D receptor is expressed on NK cells, NKT cells, γδ T-cells, and some alpha-beta (αβ) T-cells and mediates immune cell activation [141,142]. The ligands for NKG2D include MICA, MICB, and ULBP1-6, and are commonly upregulated upon cellular stress [141,143]. Therefore, many human cancers experience an upregulation in NKG2D ligands, such as colorectal cancer, leukemia, lymphoma, MM, ovarian cancer, prostate cancer, and melanoma [144,145,146]. Zhang et al. [70] modified murine T-cells with a natural receptor-based chimeric protein, where the NKG2D receptor is directly connected to the intracellular CD3ζ signaling domain, and explored antitumor efficacy in a murine model of lymphoma. Indeed, NKG2D CAR T cells inhibited the growth of NKG2D ligand-positive tumor cells in vitro and in vivo. Interestingly, mice that remained tumor-free after NKG2D CAR treatment were resistant to a rechallenge of tumor cells, suggesting the development of endogenous anti-tumor immunity. This was then recapitulated in a murine model of ovarian cancer [71], and in a murine model of myeloma [72,73], with NKG2D CAR-derived IFNγ and granulocyte-macrophage colony-stimulating factor (GM-CSF) secretion important for these effects [74,75,76]. Host-specific reactions to NKG2D CAR treatment were also responsible for endogenous anti-tumor response, as enhanced CD4+ and CD8+ tumor-specific memory responses [77], increased activation of host T-cells and NK cells at the tumor site [72], and an increase in the number of tumor-specific host CD4+ T-cells accumulating in the tumor and the draining lymph nodes were observed [78]. 

Interestingly, the NKG2D CAR was also able to induce specific cytotoxicity against tumor vasculature through interaction with another NKG2D ligand, Rae1 [79]. Treatment of mice with CAR T cells resulted in tumor vasculature cell death, reduced angiogenesis, and decreased tumor growth, even in tumors that did not express NKG2D ligands. Again, a dependence on CAR T secreted IFN-γ was shown to be important. In addition, Sentman et al. [80] evaluated the potential for on-target, off-tumor toxicity in a murine model. They found that treatment of mice with high numbers of NKG2D CAR T cells resulted in enhanced IFN-γ and IL-6 secretion, like CRS shown in clinical trials. The toxicity was perforin-, GM-CSF-, and host immune system-dependent.

Human NKG2D CAR T cells were cytotoxic against NKG2D ligand-expressing tumor cells and were not impaired by soluble NKG2D ligand [81]. Together with the preclinical data from the murine models and data supporting the large-scale manufacturing of NKG2D CAR T cells [82], clinical trials were initiated. A phase I dose-escalation study evaluated the safety and efficacy of autologous NKG2D CAR T cells in patients with AML or multiple myeloma (MM) (NCT02203825). Doses of 1 × 10^6^ to 3 × 10^7^ CAR T cells were administered without dose-limiting toxicities, CRS, neurotoxicity, or autoimmune reactions [83]. Unfortunately, CAR expansion and persistence was limited. Four other clinical trials evaluated the treatment of NKG2D CAR T cells in patients with colorectal cancer (NCT03310008), AML (NCT03612739, NCT03466320), and multiple solid and hematological tumors (NCT03018405). All have either been completed, are not actively recruiting, or their status is unknown, and was sponsored by the biotechnology company, Celyad Oncology.

Recently, Celyad Oncology has optimized the NKG2D construct to include a TCR Inhibitory Molecule (TIM) that reduces endogenous TCR signaling and decreases allogenicity of the NKG2D CAR T cells (termed CYAD-02/CYAD-101). These next-generation CAR T cells are being evaluated in a phase I clinical trial using autologous CYAD-02 for the treatment of relapsed or refractory AML via single infusion following pre-conditioning chemotherapy (NCT04167696) and using allogenic CYAD-101 for the treatment of unresectable or metastatic colorectal cancer following standard chemotherapy (NCT03692429). The use of allogeneic CYAD-101 is also being evaluated in combination with pembrolizumab, the anti-PD1 monoclonal antibody therapy under the Brand name Keytruda, in a phase 1b trial for the treatment of refractory metastatic colorectal cancer patients (NCT04991948).

According to the Celyad Oncology website, the NKG2D CAR has been further optimized to include the secretion of IL-18 to promote anti-cancer activity in patients (termed CYAD-203). Currently, CYAD-203 is being evaluated in pre-IND enabling studies, with IND submission for the treatment of solid tumors planned for mid-2022.

### 5.2. IL-13Rα2 

Interleukin-13 receptor alpha-2 (IL-13Rα2), the high-affinity receptor for IL-13, is frequently overexpressed in the majority of glioblastoma multiforme (GBM) tumors in comparison to normal healthy brain tissue [147]. Kahlon et al. [98] provided proof of concept data showing that first-generation, ligand-based CAR T cells targeting the IL-13Rα2 receptor through the modified IL-13 ligand significantly kill IL-13Rα2-expressing cell lines and prolong progression-free survival in GBM xenograft murine models. The modified IL-13 ligand includes a single point-mutation that confers 50-fold higher affinity to IL-13Rα2 in addition to a 5-fold lower affinity for IL-13Rα1/IL4Rα as compared to wild-type IL-13 [148], making this mutant superior for the GBM setting to provide minimal off-target effects. 

Brown et al. [56] extended this preclinical work by conducting a first-in-human clinical trial of locally-administered IL-13Rα2-targeted CAR T cells and showed feasibility for administration without therapy-related complications. They then optimized their CAR design by incorporating a 4-1BB costimulatory domain to the construct [99]. The next-generation CAR constructs proved advantageous in comparison to the first-generation construct by exhibiting superior anti-tumor potency and prolonged survival in GBM xenograft murine models. Clinical trials testing second-generation CAR T cells to treat GBM as a single-treatment (NCT02208362) and with/without checkpoint inhibitors Nivolumab and Ipilumimab (NCT04003649), in addition to the treatment of ependymoma or medulloblastoma (NCT04661384) are currently recruiting. These trials are sponsored by City of Hope Medical Center in collaboration with the clinical-stage pharmaceutical company Mustang Bio. Importantly, City of Hope is the first institution to utilize CAR T therapy for the treatment of patients with brain tumors. Remarkably, one subject was reported to have a transient complete response to IL-13Rα2-directed CAR T cell treatment, though disease recurrence 228 days after initial CAR T treatment with diminished IL-13Rα2 expression in the tumor was reported [56,100]. This further highlights the importance of incorporating antigen escape mechanisms into CAR design, even for natural receptor- and ligand-based CAR T cell therapies.

Interestingly, treatment with IL-13Rα2-directed CAR T cells may evoke host immune responses to the tumor. In one subject, IL-13Rα2-targeted CAR T treatment altered the tumor microenvironment (TME) by activating the host immune system to target the tumor through enhanced IFN-γ signaling after CAR treatment, which subsequently resulted in subsequential complete response in the subject [101].

A phase I clinical trial open to children ages 4 to 25 for the treatment of IL-13Rα2+ recurrent or refractory GBM is currently recruiting (NCT04510051), where CAR T cells are infused directly into the ventricles of the brain (rather than intravenously). 

### 5.3. ErbB Family 

The ErbB family of receptor tyrosine kinases (RTKs) is comprised of four members: ErbB-1 (also known as EGFR), ErbB-2 (otherwise known as human epidermal growth factor receptor-2, or HER2/neu), ErbB-3 (also known as human epidermal growth factor receptor-3, or HER3), and ErbB-4 [149]. While expressed at low levels in healthy adults, overexpression of ErbB-1 and/or ErbB-2, in particular, have been associated with the development of multiple cancers, including head and neck [150,151], breast [152], lung [153], gastrointestinal tract [150], and usually indicate poor prognosis. ErbB-3 appears to have an increasingly important role in tumorigenesis [151,152].

Pharmaceuticals directed toward ErbB-1 or ErbB-2 have gained traction in the treatment of many solid tumors; however, cancer cells often acquire resistance to these therapeutics due to enhanced signaling through other non-targeted ErbB receptors [154]. Davies et al. [102] designed a ligand-based CAR that targets the entire family of ErbB receptors to circumvent these obstacles. In a preclinical study, the seven N-terminal amino acids from human transforming growth factor-alpha (TGF-α) were fused to the 48 C-terminal amino acids of epidermal growth factor (EGF) to produce the ErbB ligand coined T1E. IL-4 signaling was used to selectively expand gene-modified cells through an ectopically expressed IL-4Rα/IL-2Rβ chimeric cytokine receptor termed 4αβ [155]. Indeed, targeting of ErbB family-expressing tumor cell lines by CAR T cells resulted in enhanced killing capacity and cytokine secretion in vitro and in vivo in both head and neck squamous cell carcinoma (HNSCC) and breast cancer murine models. 

This data served as the basis for a phase I clinical trial evaluating the safety of intratumorally administered ErbB-targeted CAR T cells (termed T4 immunotherapy) in a 3+3dose-escalationn study with or without the administration of lymphodepleting chemotherapy in subjects with HNSCC (NCT01818323) [103]. To date, 16 patients have been treated with up to 1 billion CAR T cells without dose-limiting toxicities. Stable disease was achieved in 10 patients, with one patient achieving complete remission sustained over three years after treatment [104].

Safety of panErbB-targeted CAR T cells is of concern, as one patient treated intravenously with an scFv-based (not ligand-based) CAR targeting the ErbB2-receptor resulted in lethality 5 days after CAR treatment due to uncontrolled CRS [156]. Indeed, intravenously administrated T4 immunotherapy in mice resulted in lethality [105]. Despite demonstrated safety and lack of circulating panErbB-targeted cells in the peripheral blood of subjects in the ongoing clinical trial of intratumorally administered T4 immunotherapy, Kosti et al. [105] further optimized the design by investigating an oxygen-sensing system where ErbB-targeted CARs are only expressed in a hypoxic TME (termed HypoxiCAR T cells). Such a system may be incorporated as an additional safety measure in future clinical trials.

### 5.4. BCMA/TACI 

MM is the malignant transformation of plasma cells and, despite recent advances in cell therapies, remains a challenge to cure. BCMA has emerged as a promising target within the context of MM CAR T cell therapy due to high levels of expression on the terminal stages of B-cell maturation, lack of expression on HSCs, and the wide expression on nearly all cases of MM [157]. Recently, CAR T cells targeting BCMA have been approved by the FDA for the treatment of relapsed or refractory MM under the brand name Abecma (Idecabtagene vicleucel) from Celgene after the pivotal phase II KarMMa trial demonstrated substantial efficacy (NCT03361748) [9]. 

Unfortunately, loss of the BCMA antigen was suspected in a small subset of subjects treated, and there was one confirmation of biallelic genomic loss of the BCMA gene [9]. To circumvent loss of antigen expression, Lee et al. [106] hypothesized that dual-antigen targeting of BCMA and TACI by the shared antigen a proliferation-inducing ligand (APRIL) would decrease the possibility for tumor escape. Importantly, TACI is also involved in the maturation of B cells and has been shown to be expressed on MM cells [157]. 

Co-culture of BCMA/TACI-directed CAR T cells with patient-derived MM samples induced significant target cell toxicity on human MM cell lines in vitro even in the presence of soluble BCMA, TACI, or APRIL, except for the highest dose of soluble BCMA tested at 1000 ng/mL, in addition to significant IFN-γ release in comparison to control CAR T culture. Further, treatment of CAR T cells in a xenograft model of MM in mice resulted in tumor clearance. Importantly, human APRIL will bind murine BCMA and TACI with similar affinity to the human counterparts, so it was reasonable to investigate potential on-target, off-tumor toxicities associated with CAR T treatment. No treatment-related toxicities in the murine model were found. This preclinical data led to the induction of a first-in-human clinical trial [107] (NCT03287804) that was terminated in 2019 due to a lack of sufficient efficacy to warrant further investigation and increasing competition in the BCMA-targeted CAR space. Another phase I clinical trial evaluating the efficacy of BCMA/TACI-targeted CAR T cells is currently recruiting (NCT04657861). 

Schmidts et al. [57] proposed an optimized BCMA/TACI-targeted CAR design by utilizing the natural trimeric conformation of APRIL and hypothesizing this will lead to improved binding kinetics and, therefore, efficacy in MM. Indeed, optimized trimeric BCMA/TACI-directed CAR T cells (TriPRIL CAR T cells) yielded enhanced specific cell lysis and robust degranulation when cultured with BCMA/TACI+ cell lines. In a high-tumor-burden xenograft mouse model of MM, TriPRIL CAR T cells fully eradicated MM disease, whereas the original, non-trimeric CAR T cells were only able to stabilize the disease. Further, the TriPRIL CAR T cells were able to eradicate BCMA-knock-out MM cells in comparison to treatment with traditional BCMA CAR T cells and the non-trimeric CAR T cells, further demonstrating the importance of dual-targeting of TACI in the MM setting. This preclinical data forms the basis of a first-in-human clinical trial evaluating TriPRIL CAR T cells in MM patients with relapsed or refractory disease and is currently recruiting (NCT05020444). 

### 5.5. MMP2+ GBM 

Chlorotoxin (CTLX) is a small peptide isolated from the venom of the deathstalker scorpion *Leiurus quinquestriatus* that has been shown to bind to GBM tumors [158,159]. Importantly, CTLX on its own is nontoxic to both tumor cells and normal tissue [160,161,162], making it ideal for incorporating into a therapeutic to treat GBM tumors with minimal off-target toxicity. Indeed, phase I trials evaluating the safety of administering chlorotoxin-based therapeutics in adults with recurrent high-grade gliomas showed only minor adverse events and no dose-limiting toxicities [160].

Wang et al. [108] demonstrated that CTLX binds to tumor samples from 15 different patients, and co-culture of GBM cell lines with CTLX CAR T cells induces significant GBM cell killing, T-cell activation measured by upregulation of CD69 and 4-1BB, and T-cell degranulation measured by an increase in expression of CD107a. In a GBM xenograft mouse model using two different patient-derived GBM cell lines, treatment with CAR T cells induced tumor regression and prolonged survival of mice in comparison to treatment with control CAR T cells. 

CTLX has been reported to associate with many membrane proteins, including matrix metalloproteinase 2 (MMP2) [163]. Indeed, while soluble MMP2 neither activates nor inhibits CTLX CAR T efficacy, membrane-bound MMP2 expression on tumor cells is needed for successful CTLX CAR T targeting of GBM tumors [108].

This preclinical data served as the basis for a first-in-human phase I clinical trial of CTLX CAR T cell treatment in patients with MMP2+ recurrent or progressive GBM, sponsored by City of Hope Medical Center, Duarte California, USA (NCT04214392). In 2020, City of Hope licensed the intellectual property to the biotechnology company Chimeric Therapeutics Limited. As of August 2021, Chimeric Therapeutics announced Investigational New Drug (IND) clearance from the US FDA for the CTLX-CAR T cells—which will allow the clinical trial to expand to multiple sites—and announced plans for a phase I trial to investigate CTLX- CAR T cell treatment in various solid tumors. Chimeric Therapeutics has indeed already partnered with OncoBay Clinical to provide research support to expand clinical development.

Ding et al. [97] explored another CTLX-mediated CAR T cell product co-transduced with O-6-Methylguanine-DNA Methyltransferase (MGMT) to provide CAR T resistance to alkylating chemotherapeutics commonly used in cancer treatment. They show these MGMT-modified CTLX-CAR T cells elicit enhanced cytotoxicity against tumor cells in the presence of temozolomide (TMX) in vitro.

### 5.6. ICAM-1

Intracellular adhesion molecule 1 (ICAM-1) is commonly upregulated on many solid tumors, in addition to healthy endothelial cells, epithelial cells, and immune cells [164]. Park et al. [109] demonstrated that ICAM-1-targeted CAR T cells may prove advantageous for the treatment of ICAM-1+ solid tumors in preclinical models utilizing the inserted domain of lymphocyte function-associated antigen (LFA-1). This, in combination with an investigation by Vedvyas et al. [110], shows that ICAM-1-directed CAR T cells can selectively kill ICAM-1+ tumor cell lines and in preclinical animal models. Importantly, ICAM-1-targeted CAR T cells co-express the somatostatin receptor subtype 2 (SSTR2), which allows for in vivo CAR T imaging. ICAM-1 directed CAR T cells have also been evaluated preclinically in the setting of gastric cancers [111].

This data serves as the basis for a multi-center phase I clinical trial sponsored by AffyImmune Therapeutics in collaboration with Weill Medical College of Cornell University, evaluating the treatment of relapsed or refractory thyroid cancer with ICAM-1-directed CAR T cells (AIC100) (NCT04420754) and is actively recruiting. As of May 2021, the FDA has awarded AIC100 Fast Track designation.

### 5.7. FSHR 

The follicle-stimulating hormone (FSH) receptor (FSHR) is predominantly expressed on granulosa cells in the ovary and Sertoli cells in the testis [165]. Urbanska et al. [112] hypothesized that targeting FSHR in cancerous tissue, FSHR+ tumor vasculature, and FSHR+ reproductive tissue that is otherwise healthy yet non-essential for patient survival will create an attractive avenue for CAR therapy. FSHR-targeted CAR T cells utilize full-length FSH as the antigen-binding moiety and make up a new class of CAR molecules known as chimeric endocrine receptor (CER) T-cell therapy. In preclinical studies, it was found that FSHR-targeted CAR T cells specifically kill FSHR+ ovarian cancer cells in vitro and in human ovarian cancer xenograft murine models and patient-derived FSHR+ xenograft models [112,113]. Interestingly, mice treated with murine-specific FSHR-targeted CAR T cell boosted endogenous pre-existing anti-tumor immunity in an immune-competent mouse model [113], further validating results seen by Alizadeh et al. in the context of IL-13Rα2-targeted CAR T treatment in patients with GBM [101]. The authors hypothesized that this antigen-non-specific response may delay tumor progression in the event of loss of target antigen. Importantly, Anixa Biosciences Inc., the biotechnology company developing the FSHR-targeted CAR T cell therapy in partnership with the Moffitt Cancer Center, announced in August 2021 that the U.S. FDA had cleared the Investigational New Drug (IND) application, and human clinical trials may begin.

### 5.8. CD70

CD70 expression is normally restricted to a small subset of highly activated T-cells, B cells, and dendritic cells (DCs) [166,167,168]; however, CD70 has been found to be upregulated in large B-cell leukemia and lymphoma [169], T-cell leukemia [170], renal cell carcinoma [171], and glioblastoma [172]. Shaffer et al. [84] developed a natural receptor-based CAR to target CD70 by incorporating its full length receptor-ligand CD27 as the antigen-binding domain and fusing it to the intracellular signaling domain CD3ζ in a first-generation CAR construct. CD27 is expressed on T-cells and can serve as costimulation [167], so incorporating the full-length receptor into the CAR construct allows CD27 to retain its endogenous function. Indeed, co-culture of CAR T cells with CD70+ cell lines resulted in enhanced T-cell activation, cytokine release, tumor cell cytotoxicity. Additionally, CD70-targeted CAR T cells elicited significant antitumor activity in a murine xenograft model of lymphoma. Further, a comparison of the efficacy of natural receptor-based CD70-targeted CAR T cells and scFv-based CD70-targeted CAR T cells showed that the natural receptor-based CARs demonstrated enhanced proliferative potential and antitumor activity in vitro and in murine models [85].

Wang et al. [86] optimized this model by incorporating additional costimulatory domains 4-1BB and CD28 and truncating the CD27 receptor. They found that CAR T cells extended the survival and decreased tumor size in a xenograft model of renal cell carcinoma. Importantly, they did not encounter any fratricide toward CD70-expressing activated T-cells during the expansion process or in vivo studies. A phase I/II clinical trial evaluating the efficacy of CD70-targeted CAR T cells containing the truncated form of CD27 and the intracellular costimulatory domain of 4-1BB in patients with CD70+ pancreatic cancer, renal cell cancer, breast cancer, melanoma, and ovarian cancer has been initiated, but is currently suspended (NCT02830724).

### 5.9. GMR

The GM-CSF receptor (GMR) is comprised of two subunits: the α subunit (CD116) and a common β subunit (CD131). The α subunit, CD116, is highly expressed in AML, normal myeloid cells, and a rare subtype of AML known as juvenile myelomonocytic leukemia (JMML) [173]. These observations have led to the production of a ligand-based CAR in which the GM-CSF protein is utilized as a binding domain to target CD116+ cells [62,114]. Indeed, in vitro co-culture of CAR T cells with CD116+ cell lines and primary JMML patient samples induced significant cytotoxicity, while co-culture with cord blood and bone marrow from a healthy donor did not. 

Unwanted signaling of GM-CSF through targeting the GMR is a problem of concern as this can lead to tumor growth upon CAR treatment. Interestingly, Lopez et al. [174] and Hercus et al. [175] demonstrated that GM-CSF mutations at residue 21 fail to agonistically signal through the GMR, in addition to the decreased binding ability to high-affinity GMR receptors while maintaining the equivalent binding ability to those of low-affinity. As stated previously, tailoring the affinity of each CAR construct is important for optimal in vivo functionality [60,61]. Therefore, to decrease the chance of unwanted receptor signaling and enhance anti-tumor efficacy, GMR CAR T cells were optimized by replacing the native GM-CSF protein with an E21K GM-CSF mutant to improve long-term in vitro anti-tumor activities. Indeed it was shown that optimized GMR CAR T cells do elicit enhanced anti-tumor responses both in vitro and in murine models with disseminated AML [62]. The GMR CAR T cell therapeutic is currently being investigated in an investigator-initiated, first-in-human trial currently recruiting in Japan (jRCT2033210029).

### 5.10. Antibody-Coupled T-cell Receptor (ACTR) 

Antibody-dependent cellular cytotoxicity (ADCC) is the process by which innate immune cells recognize target cells opsonized by monoclonal antibodies through their Fc receptors and trigger target cell death [176]. The FcγRIIIA receptor, or CD16, is utilized by NK and γδ T-cells to mediate ADCC through the binding of Fc regions of monoclonal antibodies [177]. Therefore, many preclinical studies evaluated the effect of fusing the ectodomain of CD16 with the intracellular domain of FcεRIγ [87], the intracellular domain of CD3ζ [88], or the intracellular signaling domains of 4-1BB [89] or CD28 [90] and CD3ζ to produce a natural receptor-based CAR. All showed promising results of enhancing the cytotoxic potential of T-cells when co-cultured with monoclonal antibodies, specifically with anti-CD20 monoclonal antibody rituximab on leukemia cell lines, anti-HER2 monoclonal antibody trastuzumab on breast and gastric cancer cell lines, and anti-GD2 monoclonal antibodies on neuroblastoma and OS cell lines [89].

This preclinical success prompted many clinical trials evaluating Fc-targeted CAR T cells, otherwise known as Antibody-Coupled T-cell Receptors (ACTRs), all sponsored by the biotechnology company Cogent Biosciences (previously known as Unum Therapeutics). The first was a phase I, multi-center clinical trial evaluating the efficacy of ACTR containing the 4-1BB costimulatory domain (ACTR087) in patients with relapsed or refractory CD20+ B-cell lymphoma in combination with rituximab (NCT02776813). Unfortunately, the FDA placed the clinical trial on hold after one subject died from neurotoxicity and two subjects died from sepsis after suffering CRS. After it resumed, the trial was again placed put on hold because of grade 3 neurotoxicity, grade 4 respiratory distress, and cytomegalovirus (CMV) infection. This trial has since been completed, but results have not been posted. Other clinical trials utilizing ACTR087 include a phase I study evaluating the combination of ACTR087 with a monoclonal antibody targeting BCMA (NCT03266692) and a phase I study in HER2+ solid tumor cancers with the monoclonal antibody trastuzumab (NCT03680560). It appears these trials have been terminated to shift focus to a new design.

ACTR707 is the next-generation version of ACTR087, replacing the 4-1BB costimulatory domain with a CD28 costimulatory domain. Clinical trials evaluating ACTR707 include a phase I study in HER2+ solid tumors (NCT03680560) and a phase I study in combination with rituximab in patients with a relapse of refractory B-cell lymphoma (NCT03189836). Unfortunately, the FDA placed another hold on ACTR707 after a report of one subject who developed a potential new malignancy possibly related to CAR treatment. All clinical trials have been terminated by the sponsoring company. Further, a prospective long-term follow-up study of all subjects treated with an ACTR product for safety monitoring began (NCT02840110), though it is now also terminated. These trials may have been ill-fated from the start due to competing serum immunoglobulins, as was the case with anti-IGK CARs in the context of MM [178].

## 6. Summary

Although little is known about the biology of some natural receptor-based and ligand-based CAR constructs, some have advanced to clinical testing and have shown some promising results. However, as discussed, additional research is needed to fully understand the best applications for these emerging and evolving technologies. It is also clear that there are opportunities where natural receptor- and ligand-based CARs offer advantages over the traditional scFv-based CAR design. Specifically, the application of natural receptor- and ligand-based CARs may provide a solution to one of the main issues of scFv-based CARs: scFv aggregation and induction of tonic CAR T cell signaling in vivo. However, it cannot go unsaid that these alternative designs come with their own unique challenges. For instance, natural receptor- and ligand-based CARs may have greater off-tumor toxicities due to multiple ligand-binding partners. In addition, there may be enhanced binding competition with the endogenous ligand and risk of unintentional signaling through the targeted receptor. Therefore, optimization to ensure highly functional CAR T cells is required in both designs, and the benefits of using one design over the other are highly contextually specific and require thorough investigation.

## Figures and Tables

**Figure 1 cells-11-00021-f001:**
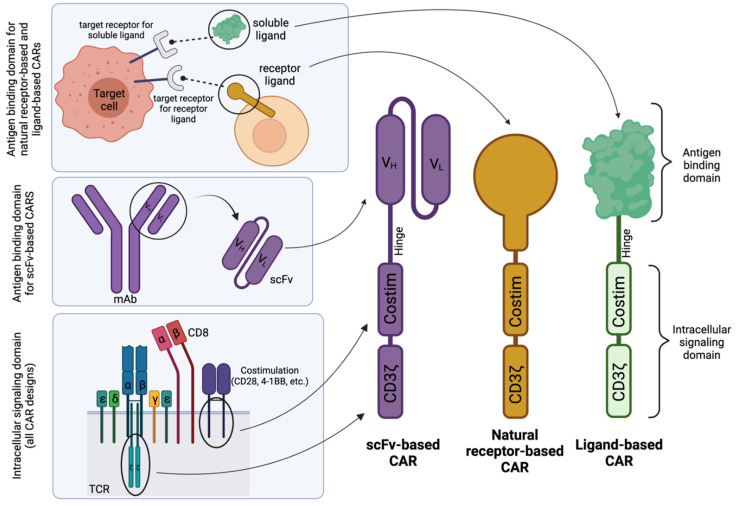
Schematic structures of scFv-based, natural receptor-based, and ligand-based CAR designs. Generally, the intracellular signaling domain is derived from components of the T-cell receptor (TCR). The antigen-binding domain for scFv-based CARs is derived from a single-chain variable fragment (scFv) from a monoclonal antibody (mAb) specific for a target receptor, whereas for natural receptor- or ligand-based CARs, the antigen-binding domain is derived from the receptor or soluble ligand whose natural binding partner is the target receptor. The complementary DNA (cDNA) that encodes the entire protein sequences of each domain or specific binding portions of the ligand or receptor are then cloned into an expression cassette for optimal CAR expression in ex vivo cultured immunocompetent cells. Costim; Costimulation. V_H_; Variable heavy. V_L_; Variable light. Created with BioRender.com.

**Figure 2 cells-11-00021-f002:**
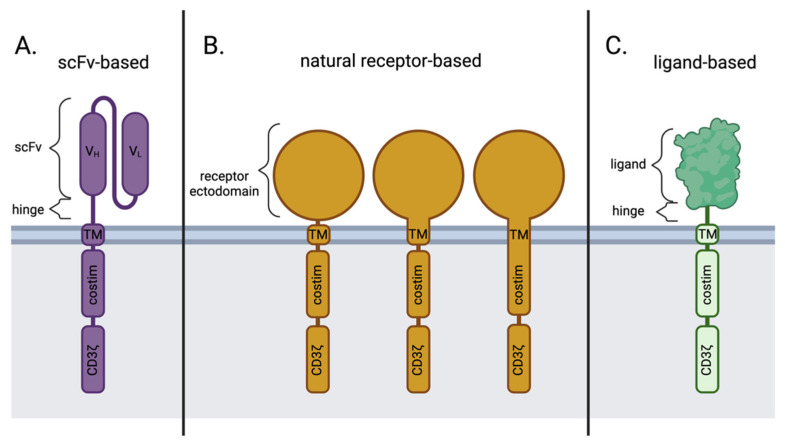
Key differences in antigen-binding domains between scFv-based, natural receptor-based, and ligand-based CAR designs. (**A**) scFv-based CARs; (**B**) natural receptor-based CARs may contain the transmembrane and/or costimulatory domains from the natural receptor itself or other sources, like CD28 or 4-1BB; (**C**) ligand-based CARs. Costim; Costimulation. scFv; single-chain variable fragment. TM; transmembrane. V_H_; Variable heavy chain. V_L_; Variable light chain. Created with BioRender.com.

**Figure 3 cells-11-00021-f003:**
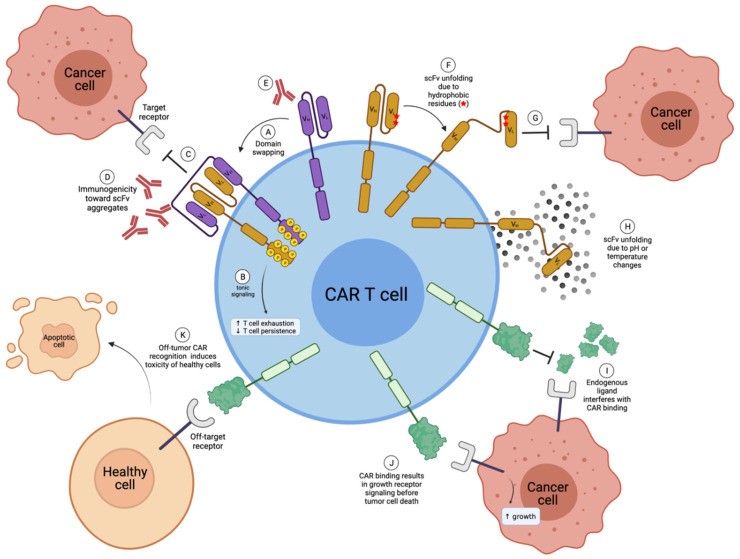
Disadvantages of scFv-based and natural receptor- or ligand-based CAR T cells. (**A**) Multiple scFvs can undergo domain swapping, where the V_H_ of one scFv will incorrectly pair with the V_L_ of another scFv, creating CAR aggregates; (**B**) CAR aggregates induce tonic signaling in CAR T cells, which is known to enhance T-cell exhaustion and decrease T-cell persistence in vivo; (**C**) scFv aggregates can no longer bind to the target receptor; (**D**) receptor aggregates can be immunogenic; (**E**) in some cases, scFvs can be immunogenic; (**F**) hydrophobic residues within scFvs can cause unfolding; (**G**) unfolded scFvs are no longer able to bind to the target receptor; (**H**) scFv unfolding can also be due to pH or temperature changes, for example during freeze-thaw cycles; (**I**) endogenous ligand (or receptors) may interfere and compete with CAR binding for the target antigen in the case of natural receptor-based or ligand-based CARs; (**J**) natural receptor- or ligand-based CAR binding may induce detrimental target receptor cell signaling; (**K**) natural receptor- and ligand-based CARs can have greater off-target toxicity. V_H_; Variable heavy chain. V_L_; Variable light chain. Created with BioRender.com.

**Table 1 cells-11-00021-t001:** Natural receptor- and ligand-based CARs are currently in preclinical and clinical development.

	Receptor Antigen	Ligand	Indication	Phase of Development	Clinical Trial	Status	Ref
Natural receptor-based CAR T cells	B7H6	NKp30	various	Preclinical			[66,67]
	DNAM-1	PVR, Nectin-2	various	Preclinical			[68,69]
	NKG2D	MICA, MICB, ULBP1-6	various	Clinical	NCT02203825 NCT03018405 NCT03310008 NCT03612739 NCT03466320 NCT04167696 NCT03692429 NCT04991948	Completed Unknown Active Withdrawn Completed Recruiting Recruiting Recruiting	[70,71,72,73,74,75,76,77,78,79,80,81,82,83]
	CD27	CD70	various	Clinical	NCT02830724	Suspended	[84,85,86]
	CD16	Fc	various	Clinical	NCT02776813 NCT03266692 NCT03189836 NCT03680560 NCT02840110	Completed Terminated Terminated Terminated Terminated	[87,88,89,90]
Ligand-based CAR T cells	EGFR	E3 Adnectin	Solid tumors	Preclinical			[91]
	FLT3	FLT3L	AML	Preclinical			[92,93]
	IL-10R	IL-10	AML	Preclinical			[94]
	MPL	TPO	AML	Preclinical			[63]
	IL-11Rα	IL-11	OS	Preclinical			[95]
	EPHB4	EPHRIN B2	RMS	Preclinical			[96]
	unknown	CTLX	GBM	Preclinical			[97]
	IL-13Rα2	E13Y IL-13	GBM	Clinical	NCT02208362 NCT04003649 NCT04510051 NCT04661384	Recruiting Recruiting Recruiting Recruiting	[56,98,99,100,101]
	ErbB family	T1E	HNSCC	Clinical	NCT01818323	Recruiting	[102,103,104,105]
	BCMA, TACI	APRIL	MM	Clinical	NCT03287804 NCT04657861	Terminated Not yet recruiting	[106,107]
	BCMA, TACI	TriPRIL	MM	Clinical	NCT05020444	Recruiting	[57]
	unknown	CTLX	GBM	Clinical	NCT04214392	Recruiting	[108]
	ICAM-I	LFA-1	Thyroid	Clinical	NCT04420754	Recruiting	[109,110,111]
	FSHR	FSH	Ovarian	Clinical			[112,113]
	GMR	GM-CSF	AML, JMML	Clinical	jRCT2033210029	Recruiting	[62,114]

## Data Availability

Not applicable.

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
