# Peer review of "Natural Receptor- and Ligand-Based Chimeric Antigen Receptors: Strategies Using Natural Ligands and Receptors for Targeted Cell Killing"

_cells, 2021, doi:10.3390/cells11010021_

Round 1

Reviewer 1 Report

Comments to Authors

The authors review previous works on the use of natural receptor(NR)- and ligand(L)-based chimeric antigen receptors as anti-cancer treatments. I found the review interesting in that it introduces a concept that might be beneficial for the development of CARs. However, in its current format the review is very long for the amount of information it contains making it hard to follow. It also gives the impression of trying to convince the readership that natural receptor and ligand-based CARs would be superior to scFv-based CARs while there is no enough evidence to make that point.

As I explain in more detail below, I think the review would greatly benefit from several changes to make it more unbiased and focused, by stripping down unnecessary details and comments, and by reducing its size to around half or two thirds of its current size.

There are several points that should be redacted for this purpose:

In sections 1 and 2, the authors enumerate several disadvantages of scFv to justify why NR and L-CARs would be advantageous, however, most of these arguments are debatable. For example, 1) antibodies are not always available, however antibodies can always be generated and this is not really a limitation of the technology; 2) scFv are not always stable in organic solvents or freeze conditions, however, this is not a condition that would ever be found in the patient; 3) scFv is prone to aggregation leading to inefficient CARs, this is the Major point that is really supported by evidence; 4) scFvs may lead to higher immunogenicity, this is true but mostly due to the murine/viral source of scFv vector and not a direct consequence of the scFv. Although the authors later cite works showing that protein aggregates may elicit immune responses it is not clear that it is what happens on CAR-T cells or if that happens how relevant it is for therapy. So, though I agree with the authors that there might be an advantage of NK and L-CARs over scFv-CARs, the major evidence for this would be the aggregation propensity of scFvs. I suggest removing the focus on the more speculative points, though they can be mentioned.

On section 2 “Advantages of natural receptor….” I found a similar issue. It really doesn’t focus on “advantages” but on the disadvantage of scFv because of their immunogenicity and aggregation. The same concept is highly repeated in almost all the subtitles. For example 2.1 scFv stability talks about hydrophobic residues in scFv that leads to aggregation, 2.2 domain swapping (which would be mediated by hydrophobic residues as well) that leads to aggregation, 2.3 scFv Immunogenecity talks about how scFv may lead to an immune response because they could be mouse derived; 2.4 Antigen escape, doesn’t talk about antigen escape but that double scFv to avoid antigen escape would aggregate more, though it is not clear if that would happen; 2.5 evolution of affinity, they claim that this would be more beneficial by using natural receptors instead of scFv, but it is not clear why that would that be the case. My suggestion is to greatly reduce this section and to point to what the real advantage would be, otherwise it gives the impression to bias.

The review continues by describing in great detail the pre-clinical and early clinical studies on NR and L CARs. These sections are very detailed in outlining what was done and would represent a good source of information. However, the sections could be reduced/reformatted to strip down unnecessary details while explaining the meaning of the findings, which are not always evident. This would facilitate the reading. Just to provide an example of how I would personally do this, I rewritten section 3.4 (lines 340-355) below reducing it from 218 words to 97 words while containing the same information:

“Chen et al targeted IL-10R in AML cells using a ligand-based CAR. Co-culture of CAR-T cells with AML cells lines led to CAR-T cell activation as shown by increased CD69 and CD25 levels, increased IFN-g, TNF-a and granzyme secretion, and elimination of AML cells in culture. This in vitro CAR-T cell efficacy correlated with increase survival in an AML xenograft model. As expected, monocytes were also eliminated due to their high IL-10R levels. The prospects of a clinical application of this CAR are highlighted by an undisrupted CD34+ HSC cell population, a condition required for T-cell therapy”

Other points that need correction:

  • Lane 309. “ immunogenicity and tonic signaling….” Immunogenicity by domain swapping, was this published?
  • Lanes 316-320 “ While further investigation…”. Delete last paragraph
  • Lane 330 and lane 345 “Enhanced degranulation via CD107a expression”. Increased cell surface CD107 is a marker of degranulation and not due to CD107a expression
  • Lanes 333-336. What is colony formation?
  • Lane 358 “ Shorter complete remission” change to “faster relapses”?
  • Lane 366-370. Claims that toxicity of CAR is good to destroy BM because they get BM transplant anyway. So why do a CAR treatment then, is the concept to replace preconditioning with CAR treatment, that should be stated if it is supported by studies?
  • Lane 379-380 “Therefore….”. Delete
  • Lane 395-396. “This widespread expression…” If it precludes CAR-T cell therapy just state it instead of “more challenging’
  • Lane 409-410. “without off target effects…”. How does weight loss indicates off target effects?, it better represents general toxicity.
  • Lane 588. “unfortunately”. Delete
  • Lane 605-608. “The clinical trial is ….” Delete, does not provide any relevant information.
  • Lane 611-612. “ via an unknown interaction with a cell receptor”. Binding to cells is enough, no need to say that there is an unknown interaction with an unknown receptor.
  • Lane 708 “…for unknown reasons”

In summary, the concept is interesting, and the review would represent a good reference.  It is clear that a great effort was put to produce this manuscript as seen by the extensive and detailed description of the bibliography and relevant citations. However, in an effort to account for every detail that was published, the manuscript ended up being extremely long, highly descriptive and with few novel concepts. There is no prioritization of what is most relevant and what is most speculative, making it extremely difficult to read. The manuscript contains approximately 800 lines of text (12000 words?) and I think it would greatly benefit to be reduced to approximately half of that.

Reviewer 2 Report

This is a very well-written and exceptionally well-illustrated comprehensive review. I strongly endorse the publication with only minor revisions necessary, most of which are optional. Below I am listing the points to be addressed in the order of how they appeared in the manuscript.

  1. Discovery of CARs: it wasn’t just Z. Eshhar’s group. I would suggest referencing at least the paper by Kuwana et al., BBRC 1987 (two years before the Gross et al. paper). If one thinks of the “classic” CAR structure as we know it today, then again the report by Eshhar et al., PNAS 1993 was contemporaneous to the paper by Brocket and Karjalainen, Eur J. Immun, 1993.

  1. “and can, therefore, elicit an immune response against cancerous or virally infected cells.”àIn my opinion, pretty much any cells, not just cancerous or virally infected, can be targeted.
  2. “Primarily, scFvs hold endless possibilities for target antigen specificity given one large stipulation: there exists a monoclonal antibody for the given receptor of choice.”

One could counterargue that the same stipulation applies to the ligand/receptor CAR: provided that there is a matching ligand/receptor. This is partially addressed below, but does not appear to be an advantage of any kind.

  1. “issues surrounding scFv immunogenicity [19] can lead to inferior in vivo function”

Please, double-check if side-by-side comparisons of such scFvs (i.e. one immunogenic and one non-immunogenic scFv of identical affinity and specificity) have actually been performed, i.e. that this is a real clinical problem in heavily pre-conditioned patients. One may wish to re-word here a little and stress that in certain cases treatment discontinuation or elimination of CAR Ts was required – which in turn MAY translate into inferior efficacy.

  1. “Drastic temperature changes are primarily of concern during the potential freeze-thaw process in CAR T development” --> Well, CAR Ts make the new, properly folded CARs after thawing and continue to do so once infused into the patient. Also, scFvs are typically selected at room temperature, with CARs being tested in vitro and in vivo at temperature close to 36.6 anyways, so this argument does not appear to be well-substantiated to me.
  2. “can also have profound effects on scFv stability”à True, but it must be kept in mind that such CAR designs are rapidly identified and eliminated at in vitro/in vivo screening steps.
  3. Tonic signaling: I wonder whether placing a tonic-signalling CAR under native TCR-like regulation would alleviate the problem? I.e. is it best to invest time in selection of the antigen-recognition module that lacks tonic signaling, or choosing the appropriate promoter is the way to go?
  4. “Importantly, soluble scFv and CAR aggregation inhibit the recognition of the target antigen”. I did not quite understand where soluble scFv comes from, unless it is some form of shedding/flawed CAR design.
  5. “In comparison, natural ligands and receptors have decreased dimerization and domain swapping risk, apart from some multimeric ligands. However, multimeric ligands pose little risk of immunogenicity as they are naturally occurring oligomerizing interactions.” I wonder whether this has ever been formally tested. It appears logical, yet somewhat speculative to me.
  6. “in decreased CAR T function and persistence [19]”à Decreased compared to what? For instance, FMC63 is fully murine – is that a terrible clinical problem for the treated patients?
  7. “In summary, treatment with antibodies derived from murine sources, like the anti-CAIX CAR T cells described above, can result in the development of human anti-mouse antibodies (HAMA), and are preferably not used in the development of clinical product candidates”à

In contrast to CAR Ts, antibodies are not necessarily given to similarly pre-conditioned patients, so the chance of anti-CAR immunity appears to be lower, right? Could you elaborate on this a little more?

  1. I am wondering whether there are receptor/ligand pairs with known allelic variants, i.e. one ligand-CAR would work well in some, but not all people – and hence be immunogenic? Could you provide any estimates here, say, by looking at 5 most advanced ligand/receptor CARs? This would strengthen the argument on the advantage of ligand-based CARs. Also, you may wish to speculate a little on the immunogenicity of (scFv/ligand/receptor)-(hinge/TM) junction.
  2. “Ghorashian et al. [59] developed a CD19-targeted scFv with greater than 40-fold lower affinity for CD19 (CAT) than the scFv commonly used in clinical trials (FMC63).” àProbably it is also worth noting that the epitopes of CAT and FMC63 are nearly the same, otherwise affinity difference would not be the sole reason for improved in vivo performance.
  3. Ease of preclinical development plans: Please consider adding that with scFv-based CARs one also has to make sure the scFv does not recognize something unintended (which the original mAb didn’t recognize, yet scFv recognizes).
  4. FLT3-targeting CARs: just a single study is referenced, there’s more.
  5. Wang et al. [164] optimizesàoptimized
  6. Antibody-Coupled T Cell Receptor (ACTR): all these trials may have been doomed for a simple reason that there are way too many competing IgGs in the serum, much as was the case with anti-Igk CARs in multiple myeloma.
  7. “While the relative affinities of natural receptors and ligands can also be tuned to opitimize CAR functionality”: Tuning affinity and signaling in natural ligand-receptor pairs may run hand in hand with immunogenicity, so this may counterweight the advantage of being natural.
  8. Scfv vs natural ligand CAR: the researcher facing the design of the new CAR does not have an infinite number of antigen-recognition modules to choose from: typically 1-2 candidates. So, when it comes to construct design, equal optimization efforts are required in the selection of hinge/tm/signaling combinations in either scenario. What’s the point then? Basically, one starts with the pathology and the very limited choice of actionable targets, rather than with the choice of scFv vs ligand/receptor as an antigen-recognition module.

Round 2

Reviewer 1 Report

The authors have modified the manuscript substantially making it better structured, more conceptual and objective. Therefore an easier read. It is a valuable resource for the field.

I personally do not like the summary as it is not informative, and it should say something even if you didn't read the review. I suggest to write a line or two about what the advantages and disadvantages of the technology are before stating what the possible clinical future might be.

Author Response

Dear Editors and Reviewers,

We have addressed the comment made by the reviewers. To remain consistent with our previous responses, the reviewer’s comment has been assigned a unique number (e.g. 1.1 for the first comment from Reviewer 1) and our tracked changes file notes where the critiques have been addressed.

Of importance to note, the line values of the tracked changes document are not accurate. When all changes have been accepted, the line values are correct. Our reference to line numbers to refer the Reviewers to the changes made are in reference to the tracked-changes document, not the document where the changes have been accepted.

_____________________________________________________________________________________

Reviewer #1:

The authors have modified the manuscript substantially making it better structured, more conceptual and objective. Therefore an easier read. It is a valuable resource for the field.

1.1. I personally do not like the summary as it is not informative, and it should say something even if you didn't read the review. I suggest to write a line or two about what the advantages and disadvantages of the technology are before stating what the possible clinical future might be.

Response: We thank the reviewer for this comment, and we agree. We, therefore, reformatted the Summary and added additional text describing the advantages and disadvantages of natural receptor- and ligand-based CARs in comparison to traditional scfv-based CARs (lines 764-783 of the tracked changes document).